



# Evaluating future hydrological changes in China under climate change
**Danyang Gao[1, *], Albert S. Chen[1], Toby Richard Marthews[2], and Fayyaz Ali Memon[1]**
[1] Centre for Water Systems, University of Exeter, Exeter, EX4 4QF, UK
[2] UK Centre for Ecology & Hydrology (UKCEH), Wallingford OX10 8BB, UK
[*] Corresponding Author
Corresponding author email: dg442@exeter.ac.uk



**Abstract**
Projecting and understanding future hydrological changes in China are critical for effective water resource
management and adaptation planning in response to climate variability. However, few studies have
investigated runoff variability and flood and drought risks under climate change scenarios for the entire
region of China at high resolution. In this study, we use the Joint UK Land Environment Simulator (JULES),
specifically tailored for simulating hydrological processes in China at a 0.25-degree resolution. Downscaled
and bias-corrected forcing data from Global Climate Models (GCMs), using the bias-correction and spatial
disaggregation (BCSD) method, were used to drive the JULES model to project future hydrological
processes under medium (SSP245) and high (SSP585) emission scenarios. The results indicate that annual
runoff in China is projected to increase significantly under the high emission scenario, notably in the eastern
and southern basins. Wetter summers and drier winters are expected in the south, while the opposite trend is
expected in the north. Wetter conditions in the near future and drier summers in the far future are expected
in northern China. Shifts from drier to wetter conditions are projected in the southeast and southwest areas,
while the middle Yangtze River basin may experience the opposite trend. The flood risk is expected to
increase in spring, summer, and autumn, along with heightened drought risk in winter, summer, and autumn.
Southern China would face greater flood risk, while the central Yangtze River basin would face intensified
drought risk, especially in the far future. These findings underscore the influence of different emission
scenarios on flood and drought risks, emphasizing the need for proactive measures to enhance climate
adaptation in the future.
**Keywords:** Hydrological simulation; Extreme hydrological risk; Land surface model; Climate change;
CMIP6



## 1 Introduction

Ongoing global warming is now having significant impacts on the hydrological cycle of many global ecosystems (Yin et al., 2018; Zhang et al., 2018; IPCC, 2023). Changes in the timing, magnitude, and seasonality of runoff may cause drought and flooding, posing threats to water security, which will lead to negative impacts on ecology, society and economy (Schewe et al., 2014; Miller et al., 2021). Therefore, the analysis of runoff responses to climate change is essential for investigating water security and extreme disaster events. Particularly in China, one of the most water-stressed nations (Zhai et al., 2022) with significant difference in regional precipitation (Jin et al., 2021), investigating the climate change impacts on runoff is essential for national and regional planning and the sustainable development of water resources.

Many scholars used different methods (e.g., hydrological models and the climatic elasticity methods) to study runoff under climate change in different regions of China. Zhao et al. (2019) developed an extended Variable Infiltration Capacity (VIC) macroscale hydrological model (named VIC-CAS) to project future changes in runoff components on the Tibetan Plateau based on Global Climate Models (GCMs) from the Coupled Model Inter-comparison Project Phase 5 (CMIP5). Gu et al. (2020) employed four lumped conceptual hydrological models to simulate runoff based on 31 GCMs from CMIP5 in 151 catchments in China. Guan et al. (2021) simulated the runoff conditions during the rest of 21 century based on Budyko framework and GCMs from the Coupled Model Inter-comparison Project Phase 6 (CMIP6) in 10 major river zones in China. Jin et al. (2021) assessed future water resource changes of the source region of the Yangtze River by the Soil and Water Assessment Tool (SWAT) and meteorological data of GCMs integrated by deep learning. Zhou et al. (2023a) analysed annual total runoff in 11 major basins in China by VIC. However, the model simulations in these studies were at the catchment scale that did not cover the whole region of China, while there are 2,221 rivers with catchment areas exceeding 1000 km$^2$ in China (Ministry of Water Resources,

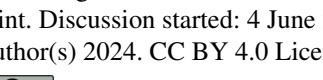



P. R. China and National Bureau of Statistics, P. R. China, 2013). Considering the numerous river basins
(Fig. 1), the difficulty of obtaining hydrologic data in China (Lin et al., 2023) and the rare observation sites
in some regions (e.g., high mountains), it is extremely difficult to calibrate and validate models for all
catchments in China.
Some global studies related to future runoff under climate change covered China region (Cook et al.,
2020; Chai et al., 2021; Hou et al., 2022; Wang et al., 2022; Miao et al., 2023). However, their analysis
mainly based on results from CMIP5, CMIP6, Inter-Sectoral Impact Model Inter-Comparison Project
(ISMIP2a) and Global Land Data Assimilation System (GLDAS), the resolutions of their runoff projections
were coarse. Besides, the results were discussed mainly on the continental scale, the specific environment
attributes of China were not particularly addressed. In this study, we will consider the features in the
application the Joint UK Land Environment Simulator (JULES) to simulate hydrological processes with high
resolution (0.25°) at the national scale.
The JULES model was developed by the UK Met Office evolved from the Met Office Surface Exchange
Scheme (MOSES, Cox et al., 1999), which was the land surface scheme of UK Met Office Earth System
Model, now used as a standalone land surface model to simulate the carbon fluxes (Clark et al., 2011), water,
energy, and momentum (Best et al., 2011) between the land surface and the atmosphere. The model has been
increasingly used for hydrological assessment (Zulkafli et al., 2013; Le Vine et al., 2016; Martínezde la Torre
et al., 2019; Yang et al., 2019; Chou et al., 2022). However, the JULES model is rarely used in China,
especially for hydrological simulation. Its ability to simulate hydrological process in China has yet to be
examined.
In this study, we ask the following questions: (1) How well can the JULES model simulate hydrological
processes in China at 0.25∘ resolution? (2) What will be the future runoff magnitude, year-to-year (inter-





annual) variability and distribution in China? (3) whether and where will China face extreme runoff hazards
risks (drought and flooding) under climate change? Addressing these questions will be crucial in enhancing
our understanding of hydrological dynamics in China and in formulating effective adaptation and mitigation
strategies to mitigate the impacts of changing climate conditions on water resources management and
disaster risk reduction.

**2 Methods**
2.2 Historical Simulation using the JULES Model

Input for the JULES model includes meteorological forcing data and ancillary data. In its standard

configuration, JULES recognises nine land cover types: broadleaf trees, needleleaf trees, $C_3$ (temperate)
grass, $C_4$ (tropical) grass, shrubs, urban, inland water, bare soil and ice (Best et al., 2011). In this study,
historical meteorological forcing data include near surface temperature, precipitation, downward shortwave
and longwave radiation, wind speed, specific humidity and surface pressure are from the European Centre
for Medium-Range Weather Forecasts Reanalysis 5 (ERA5, Hersbach et al., 2020). The ancillary data are
from Marthews et al. (2022), considering nine land cover and seven soil layers.

To generate a reasonable initial condition, the JULES model was spun up in December of 1959 with

200 spin up cycles. Main run was during 1960 to 2014 covering all of China, using 0.25° resolution and a
daily timestep. Observed discharge from the Global Runoff Data Centre (GRDC) was used to do monthly
calibration and validation (Fig. 1). The model calibration was from 1962 to 1977, 1978 to 1986 were used
for validation. The Pearson correlation coefficient ($r$) and Nash–Sutcliffe efficiency coefficient (NSE, Nash
and Sutcliffe, 1970) were used to evaluate the model performance. The equations of r and NSE are shown
in Eq. 1 and 2. Typically, an NSE greater than 0.5 indicates good alignment. Detailed standard thresholds



for NSE are provided in Marthews et al. (2022).

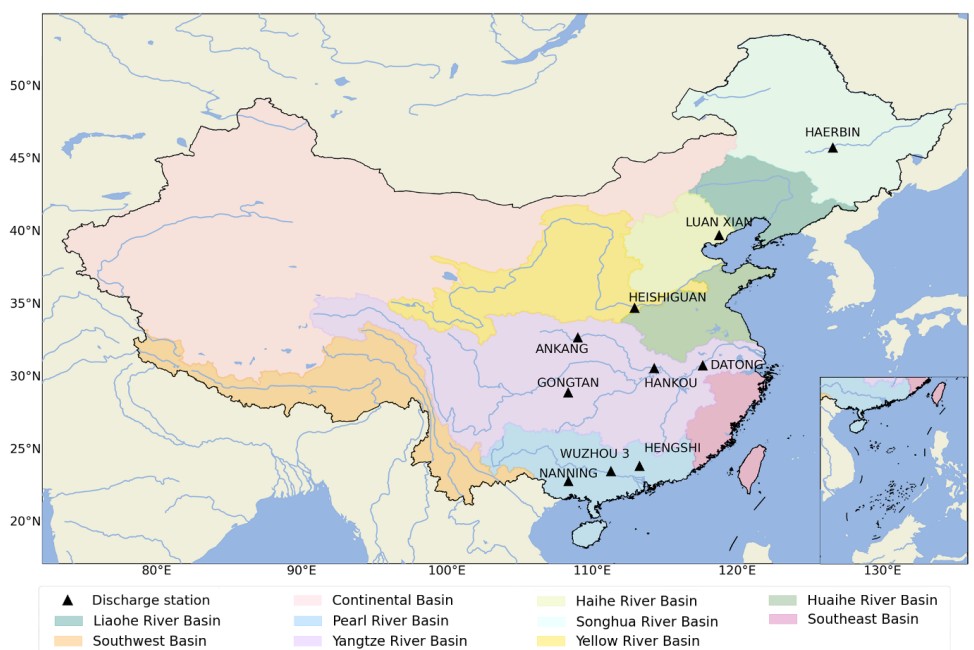

Figure 1. Location of GRDC stations for calibration and validation.

$$r = \frac{n \sum Q_o Q_m - (\sum Q_o)(\sum Q_m)}{\sqrt{[n \sum (Q_O)^2 - (\sum Q_O)^2][n \sum (Q_m)^2 - (\sum Q_m)^2]}} \quad (1)$$

$$\text{NSE} = 1 - \frac{\sum_{t=1}^{T}(Q_o^t - Q_m^t)^2}{\sum_{t=1}^{T}(Q_o^t - \bar{Q}_o)^2} \quad (2)$$

where $Q_m$ is modelled discharge, $Q_o$ is observed discharge, $\bar{Q}_o$ is the mean of observed discharges, t is
time, and n is the number of observations available for analysis.
2.2 Bias-Correction Spatial Disaggregation (BCSD) method

Future meteorological driving data are from GCMs in CMIP6. It was downscaled to 0.25° resolution

based on ERA5 by bias-correction and spatial disaggregation (BCSD) method (Wood et al., 2004; Thrasher
et al., 2022). This method compares the original GCMs output with climate observations during a common
historical reference period, and uses the information obtained from the comparison to adjust future



projections of GCMs, aiming to align the GCMs more closely with historical observation data and enhance

their realism within the specific spatial area (Thrasher et al., 2022).

The BCSD method consists of three steps: preprocessing, bias correction, and spatial disaggregation.

Preprocessing is only for the temperature variable; the main purpose is to detrend temperature so that their

climate trends would not be affected by the bias correction. The 9-year moving average is calculated in each

month individually. These trends are preserved and then re-incorporated into the adjusted data following the

bias correction process. The bias correction process corrects the bias in GCMs output by observations, firstly,

ERA5 datasets were interpolated to match the resolution of the selected GCMs. The data within ±15-day

window from GCMs and ERA5 in a reference period from 1959 to 2014 were chosen to generate two

cumulative distribution functions (CDFs). The quantile corresponding to each original GCM value was

derived from the GCM-based CDF distribution for that particular day. Subsequently, this quantile was used

to calculate the corresponding value from the ERA5-based CDF distribution. The final value is the bias-

corrected GCMs data. Spatial disaggregation process interpolates the bias-corrected GCMs data to the

observational resolution (0.25°). A smoothed daily climatology was generated over the reference period

based on ERA5 by a Fast Fourier Transform retaining three harmonics. This climatology was then

interpolated to the original grid of the GCMs and factored out of the bias-corrected GCMs either by

subtracting from the temperature variables or by dividing from the other variables. The residual fields were

bilinearly interpolated to the original 0.25° grid of the ERA5. Subsequently, the 0.25° climatology was

factored back in either through addition to the temperature variables or multiplication by the other variables,

yielding the final downscaled GCMs data.

We selected six GCMs from CMIP6 (EC-Earth3, INM-CM5-0, MIROC6, MPI-ESM1-2-HR, MRI-

ESM2-0 and NorESM2-LM, shown in Table 1) that perform well for precipitation and temperature in China





(Yang et al.,2021; Lu et al., 2022; Jia et al., 2023). First, we downscaled the precipitation of these six GCMs.
By comparing the temporal root-mean-square error (RMSE) of annual precipitation in reference period over
China, spatial Pearson correlation coefficient (*r*) and RMSE for multi-year average (1959–2014) daily
precipitation, the three best performing GCMs were selected. Then we downscaled the near surface
temperature, precipitation, downward shortwave and longwave radiation, wind speed, specific humidity and
surface pressure of these three GCMs in middle and high emission scenarios (SSP245 and SSP585) as future
input forcing data for JULES model.
Table 1. List of six CMIP6 GCMs and their reporting institutions and countries, and horizontal resolutions

| GCM name | Modelling centre/Nation | Horizontal resolution in the standard configuration |
| --- | --- | --- |
| EC-Earth3 | EC-Earth consortium / Europe | 0.703° × 0.703° |
| INM-CM5-0 | Institute for Numerical Mathematics, Russian Academy of Science / Russia | 1.5° × 2° |
| MIROC6 | Atmosphere and Ocean Research Institute, Centre for Climate System Research - National Institute for Environmental Studies and Atmosphere and Ocean Research Institute / Japan | 1.4° × 1.4° |
| MPI-ESM1-2-HR | Max Planck Institute for Meteorology /Germany | 0.9375° × 0.9375° |
| MRI-ESM2-0 | Meteorological Research Institute / Japan | 1.125° × 1.125° |
| NorESM2-LM | Norwegian Climate Centre / Norway | 1.875° × 1.875° |

2.3 Future Projection under Climate Change

The downscaled GCMs for both the historical and future periods under the two scenarios were input

into the calibrated and validated JULES model to simulate the hydrological processes. The runoff rate output



by JULES is in units of kg·m$^{-2}$·s$^{-1}$. We estimated the runoff at different time steps, measured in mm of depth.
The annual and seasonal variations in runoff over China and its watersheds were analysed using the Mann-
Kendall test (Mann, 1945; Kendall, 1975) to assess changing trends. For multi-year average runoff variation,
the historical and future multi-year annual cycles of runoff over China and individual watersheds were
compared. Differences in historical and future multi-year monthly runoff depths under two scenarios were
calculated.
To evaluate the changes in extreme runoff under climate change, we calculated the seasonal 90th and
10th percentiles for each year based on the daily values. Regional mean values for China and individual
basins were calculated to assess the changing trends of extreme runoff. Additionally, multi-year 90th and
10th percentile runoff depths were calculated for the historical period (1975-2014), near future (2021-2060),
and far future (2061-2100) for each grid.

**3 Results**
3.1 JULES Model Evaluation
The JULES model performance for hydrology was evaluated against observed discharge. The
comparison of observations and simulations during monthly calibration and monthly validation are shown
in Fig. 2 and Fig. 3 , respectively.   The $r$ and NSE values are above 0.83 and 0.64, respectively, for
calibrations, and greater than 0.78 and 0.58, respectively, for validation, indicating that the simulation
outcomes are acceptable.
Most stations with good performance are near large rivers, indicating that the model simulates better in
large rivers. This is mainly because the resolution in this study is 0.25°, which is slightly crude for simulating
discharge of small rivers. Overall, JULES performs well in hydrological modelling.






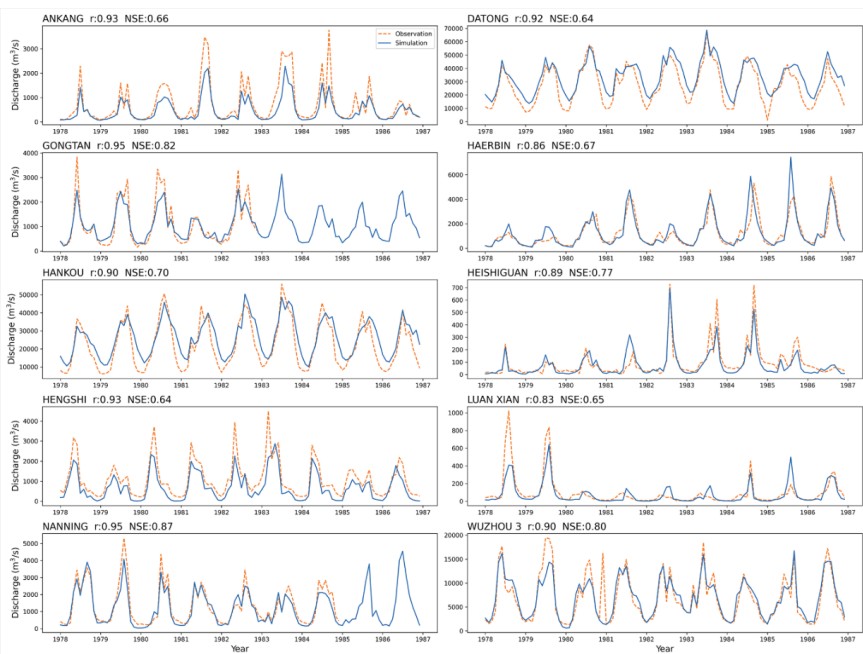

164       Figure 2. Comparison of observed and simulated discharge in monthly calibration


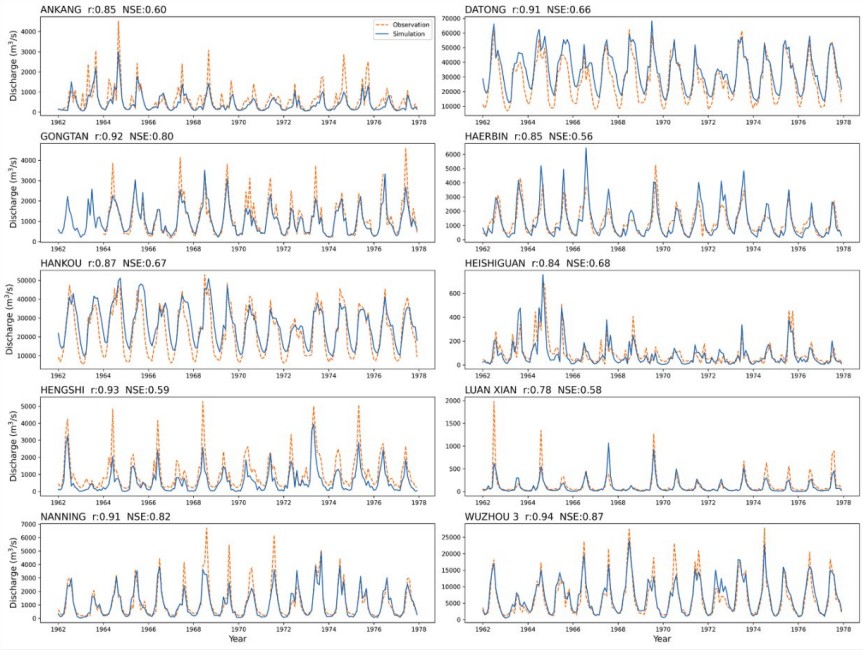

166       Figure 3. Comparison of observed and simulated discharge in monthly validation





3.2 Downscaled GCMs Evaluation

The annual precipitation from 1959 to 2014 over China of ERA5 and six downscaled GCMs is shown

in Fig. 4, while *r* and RMSE between ERA5 and each downscaled GCMs are shown in Table 2. From the
perspective of regional mean time series differences, the downscaled MPI-ESM1-2-HR performs the best.
In terms of multi-year average precipitation, the downscaled EC-Earth3 simulates the best on the spatial
pattern of precipitation (Table 2). Considering the combination of time series and spatial distribution, EC-
Earth3, MPI-ESM1-2-HR and MRI-ESM2-0 are selected.

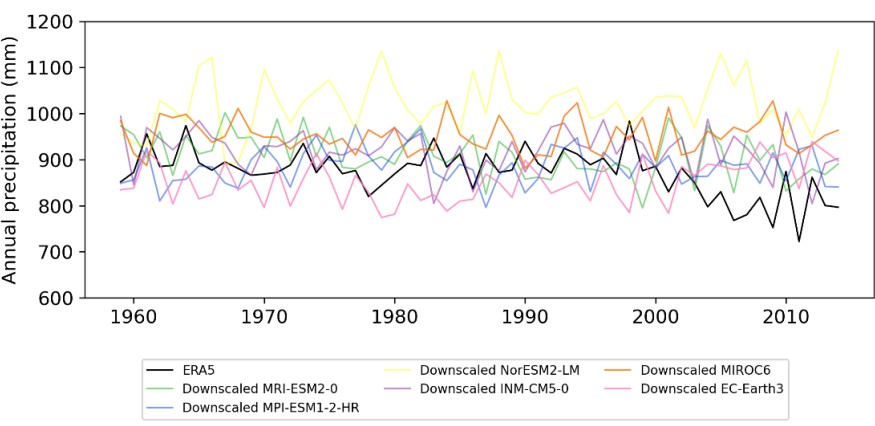


Figure 4. Comparison of annual precipitation between ERA5 and six downscaled GCMs

Table 2. *r* and RMSE for precipitation between ERA5 and six downscaled GCMs

| GCMs | RMSE with time series | *r* of spatial distribution | RMSE of spatial distribution |
|---|---|---|---|
| MPI-ESM1-2-HR | 67.057 | 0.954 | 0.819 |
| EC-Earth3 | 74.059 | 0.964 | 0.691 |
| MRI-ESM2-0 | 77.127 | 0.956 | 0.833 |
| INM-CM5-0 | 78.289 | 0.944 | 0.928 |
| MIROC6 | 104.562 | 0.941 | 1.075 |
| NorESM2-LM | 173.748 | 0.929 | 1.327 |


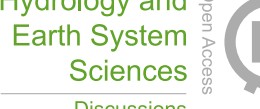



Other annual meteorological variables from 1959 to 2014 over China for ERA5 and the three selected
downscaled GCMs are shown in Fig. 5. The downscaled GCMs can simulate the trends of ERA5 for most
variables. Among them, the simulation for surface temperature is the best. A slight difference in the trend
modelling occurs in downward shortwave radiation, which is due to the original GCMs showing a clear
decreasing trend in the historical period. For all downscaled GCMs and variables except precipitation, r of
spatial distribution is greater 0.99. This indicates the selected GCMs can well reflect spatial pattern after
downscaling.

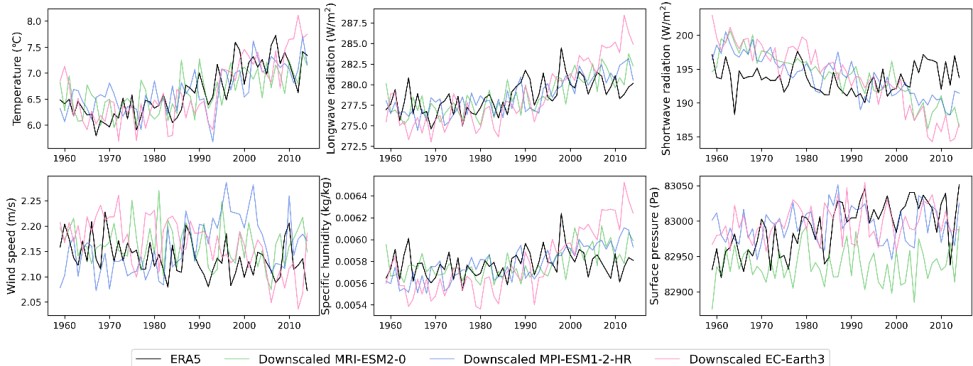


Figure 5. Comparison of annual meteorological variables between ERA5 and downscaled selected three GCMs
For each variable, to compare the bias before and after downscaling, the multi-year average values of
original GCMs and downscaled GCMs are subtracted from the corresponding values of ERA5, respectively.
The bias comparison map for every variable in ERA5 and the ensemble mean GCMs is shown in Fig. 6,
while the bias between each GCM and ERA5 is shown in Fig. S1, S2 and S3. There are obvious differences
between the original GCMs and ERA5, except for specific humidity and surface pressure. It can be clearly
seen that after downscaling through the BCSD method, the bias between GCMs and ERA5 becomes smaller.
Though the simulation for precipitation is less effective than for other variables (Fig. 6d), because
precipitation is not continuous. Therefore, this set of BSCD GCMs data can be used for JULES models
established based on ERA5 to simulate hydrological process.

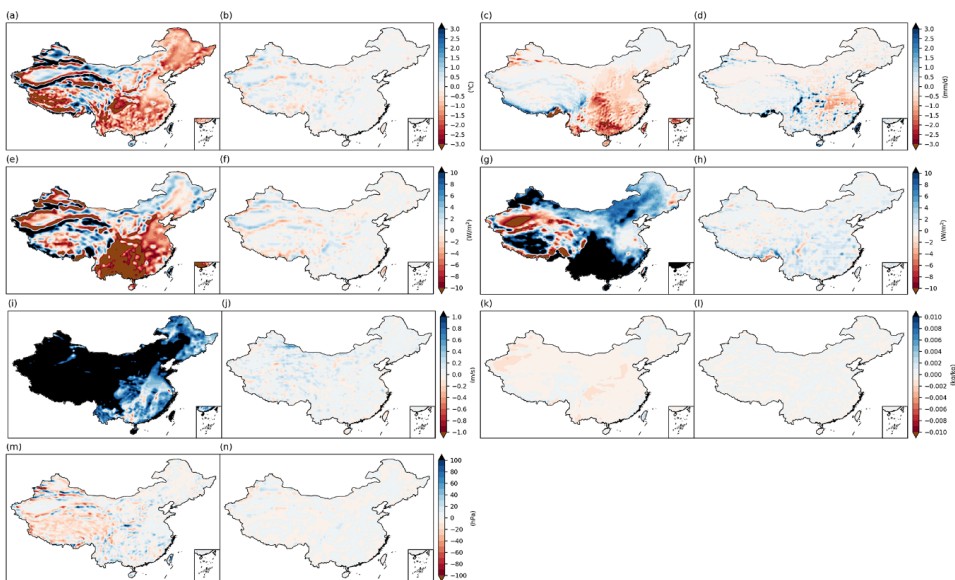


Figure 6. Bias comparison map for multi-year average daily surface temperature (a, b), precipitation (c, d), longwave
radiation (e, f), shortwave radiation (g, h), wind speed (i, j), specific humidity (k, l) and surface pressure (m, n) between
ERA5 and ensemble mean original GCMs (a, c, e, g, i, k, m), ERA5 and ensemble mean downscaled GCMs (b, d, f, h, j, l, n)
3.3 Runoff simulation results
3.3.1 Historical runoff simulation comparison

The historical runoff driven by three downscaled GCMs was simulated by the JULES model. The multi-

year (1962-2000) average daily runoff depth simulated by ERA5 subtracted from the runoff simulated by
downscaled GCMs is shown in Fig. 7. The seasonal differences are shown in Fig. 8. The difference of
simulated runoff between the three GCMs and ERA5 are basically the same. In most areas of China, there
is little difference between simulated runoff driven by GCMs and driven by ERA5. In the southeast region,
there is an overestimation of the runoff simulated by GCMs. The overestimation in the east and the
underestimation in south and middle area are more significant in summer. This is because bias of





precipitation in these areas are relatively large, especially in summer. For regional mean runoff over China,

the difference between annual runoff based on ERA5 and GCMs is insignificant (Fig. 9a). The difference in

summer varies a little more than the ones in the other three seasons (Fig. 10).

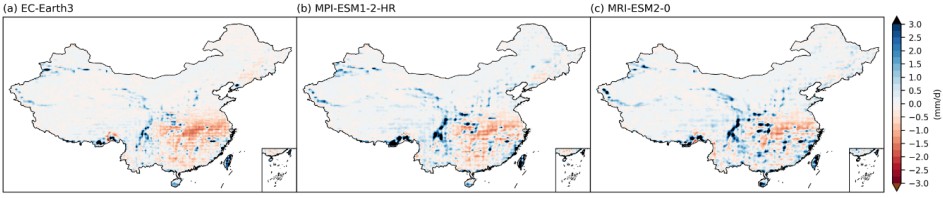

Figure 7. Multi-year (1962-2000) average daily runoff comparison map between JULES results based on ERA5 and (a) EC-

Earth3, (b) MPI-ESM1-2-HR, (c) MRI-ESM2-0.

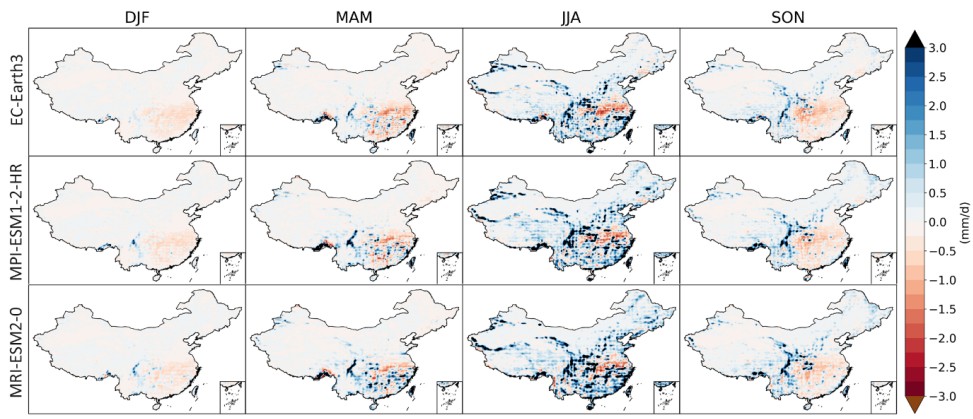

Figure 8. Multi-year (1962-2000) seasonal average daily runoff comparison map between JULES results based on ERA5 and

three GCMs.

3.3.2 Runoff variation trends

The runoff variation over China from 1962 to 2100 is shown in Fig. 9a, and the variation trends were

analysed using the Mann-Kendall test. The runoff is likely to increase significantly under the high emission

scenario, while there is no obvious trend in the historical period under SSP245. Specifically, the runoff depth

over China is projected to increase by 7.30 mm per decade between 2015 and 2100 under SSP585. This





increase is primarily attributed to the rise in precipitation. Precipitation over China is expected to increase
under both SSP245 and SSP585 scenarios (Fig. 9b). But the rising trend of runoff is not expected to be as
pronounced as that of precipitation, because the increasing trend of evaporation is expected to be more
significant in the future (Fig. 9c).
The annual runoff is likely to increase in eastern and southern China, including the Haihe River basin,
Huaihe River basin, Pearl River basin, Songhua River basin, Southeast basin and Southwest basin under
SSP585 (Fig. S4). Among these, the most dramatic increase is expected in the Southeast basin, with a trend
rate of 41.45 mm per decade (while the increase trend rate of precipitation is 59.38 mm per decade). There
are likely to be no significant trends in most basins under SSP245, with increasing trends only observed in
partial eastern watersheds (Huaihe River basin and Songhua River basin) under SSP245.

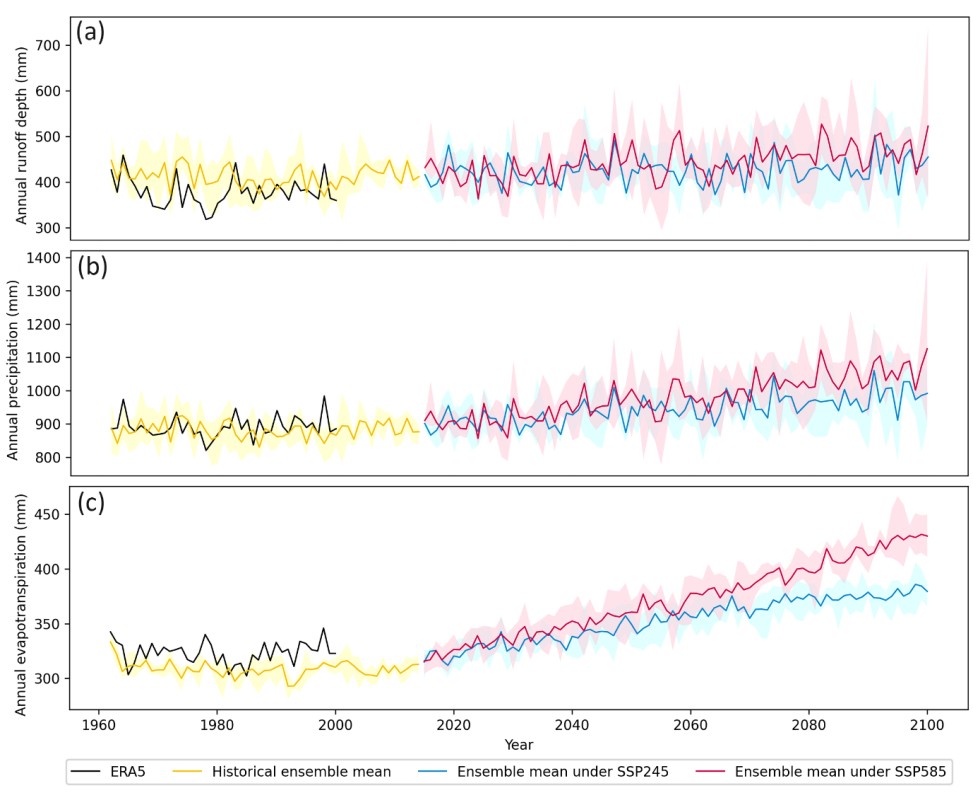






Figure 9. Annual (a) runoff depth, (b) precipitation and (c) evapotranspiration over China. The black line represents the
precipitation from ERA5, the simulated runoff and evapotranspiration based on ERA5. The yellow, blue and red lines are the
ensemble mean precipitation from the three GCMs, simulated runoff and evapotranspiration driving by the three GCMs in
historical, under SSP245 and SSP585, respectively. The shaded areas indicate the range between the maximum and
minimum values of precipitation, simulated runoff depth and evapotranspiration based on the three GCMs.
From the perspective of seasonal runoff (Fig. 10), runoff in spring shows increasing trends in the future
under both scenarios, (Fig. 10b), with the runoff depth over China likely to increase by 1.54 and 1.62 mm
per decade between 2015 and 2100. Additionally, future runoff is expected to increase in summer (Fig. 10c)
and autumn (Fig. 10d) under SSP585, with trend rates of 4.60 and 0.97 mm per decade, respectively.
The increase in runoff in each watershed is likely to occur mainly in spring and summer (Fig. S5).
However, in the Continental basin, runoff in summer is expected to decrease while winter runoff is expected
to increase both under both SSP245 and SSP585. Meanwhile, winter runoff in the Pearl River basin,
Southeast basin and Yangtze River basin is likely to show a decreasing trend under SSP585. This also
indicates that the variation trend of summer and winter runoff is likely to be opposite, with trends also
differing between the northern and southern regions of China. Southern China is expected to experience
wetter summers and drier winters under the high emission scenario, while the opposite trend is expected in
the north.



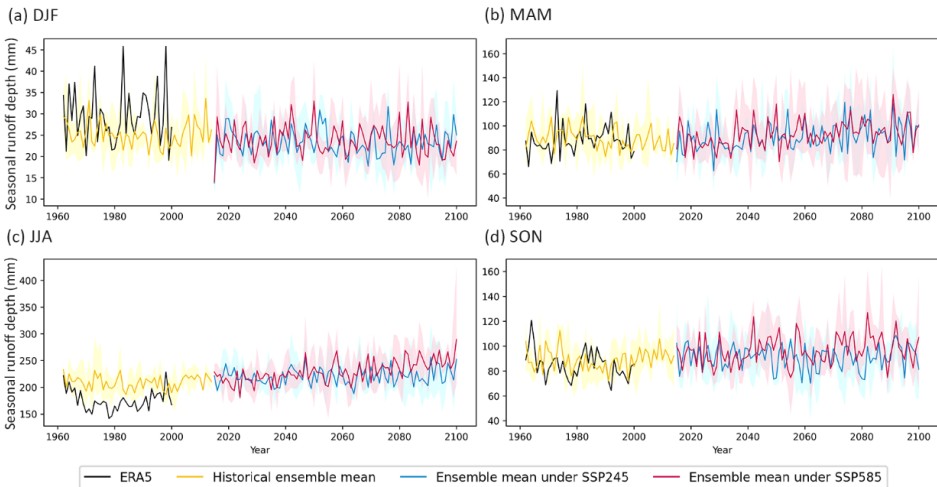


Figure 10. Seasonal runoff depth over China. The black line represents the simulated runoff based on ERA5. The yellow,

blue and red line are the ensemble mean simulated runoff driving by three GCMs in historical, under SSP245 and SSP585,

respectively. The shaded areas indicate the maximum and minimum ranges of simulated runoff depth based on three GCMs.

3.3.3 Multi-year average runoff variation

We divided future period into two parts: near future (2021-2060) and far future (2061-2100). The multi-

year annual cycle of runoff in near, far future and historical period (1975-2014) was analysed (Fig. 11).

Compared to historical period, more runoff is likely to occur in most months in the future, especially in

summer. It is expected to increase most in far future under SSP585. Similar situation shows in most

watersheds (Fig. S6). But the monthly runoff over Continental and Yellow River basin is greatest in near

future under SSP585, while it is smaller in the far future under SSP585 than it in historical period in some

summer months. This indicates wetter conditions in the near future and drier summers in the far future under

SSP585 in the northern China.

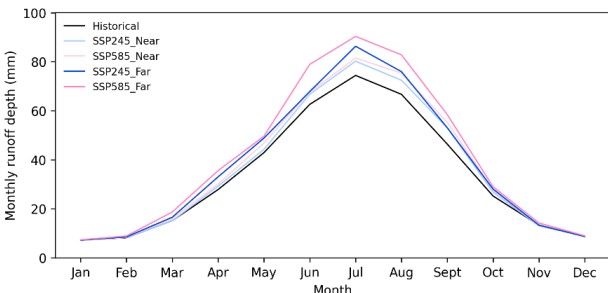


Figure 11. Multi-year annual cycle of runoff depth over China.

The multi-year monthly runoff changes in China are shown in Fig. 12. Under SSP245, the middle of

the Yangtze River basin and the southern Liaohe River basin are likely to become drier (Fig. 12a) and then
wetter (Fig. 12d). Similar trends are expected under SSP585 in the Southwest basin. Additionally, the
multi-year monthly runoff is projected to significantly increase in southern China, including the Huaihe
River, Pearl River, Southeast basins, and the eastern Yangtze River basin in the far future under SSP585
(Fig. 12e). The southeast and southwest areas are likely to become drier in the near future (Fig. 12c),
followed by wetter conditions in the far future (Fig. 12f), while the middle Yangtze River basin is expected
to experience the opposite trend.

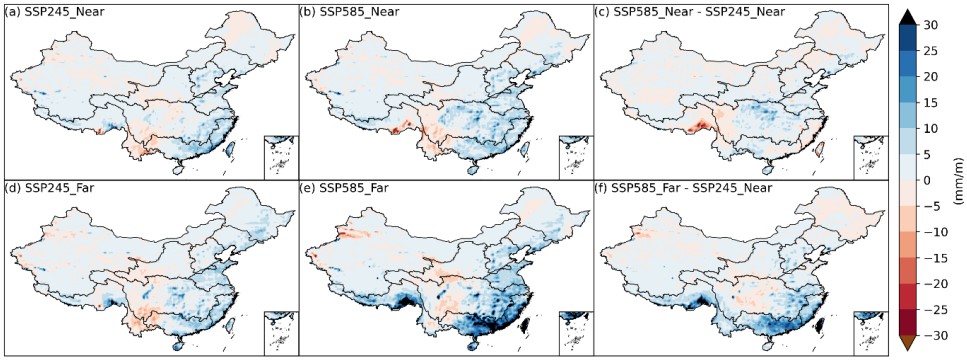


Figure 12. Multi-year monthly runoff changes in(a) 2021-2060 and (d) 2061-2100 under the SSP245, as well as (b) 2021-
2060 and (e) 2061-2100 under the SSP585, relative to the historical period (1975-2014). (c) and (f) is the difference in multi-

277        year ensemble mean monthly runoff depth between SSP245 and SSP585 in 2021-2060 and 2061-2100, respectively.



3.3 Projected Extreme Runoff Change
The ensemble mean seasonal 90th and 10th percentile runoff depths over China for each year, based on
daily values, are shown in Fig. 13. According to the Mann-Kendall test, the 90th percentile runoff is expected
to increase in spring under both scenarios, as well as in summer and autumn under SSP585. Conversely, the
10th percentile runoff is expected to decrease in winter under SSP585, in summer under SSP245, and in
autumn under both scenarios. These findings suggest an increased flooding risk in China during spring,
summer, and autumn in the future, particularly under high emission scenarios, while the risk of drought is
likely to increase in winter, summer, and autumn.

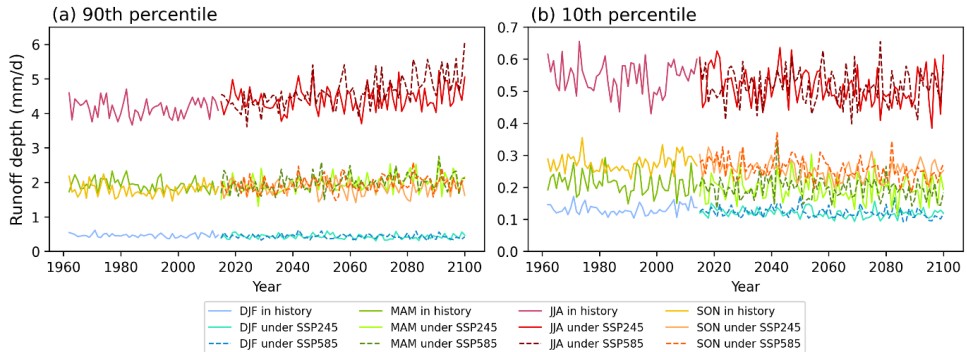


Figure 13. The ensemble mean seasonal (a) 90th and (b) 10th percentile runoff depth for each year during 1962 to 2100 over

China.

Spatial changes of the multi-year 90th percentile runoff (Fig. 14) are similar to the changes in multi-
year monthly runoff (Fig. 12). Compared to the historical period, future flood risks are likely to increase in
southern China, particularly in the Southwest basin, Southeast basin, Pearl River basin, and southern Yangtze
River basin, especially under SSP585 in the far future. In the near future, the Yangtze River basin is expected
to face a higher flood risk under SSP585 compared to SSP245, while in the far future, the flood risk in
southern China under SSP585 surpasses that under SSP245.



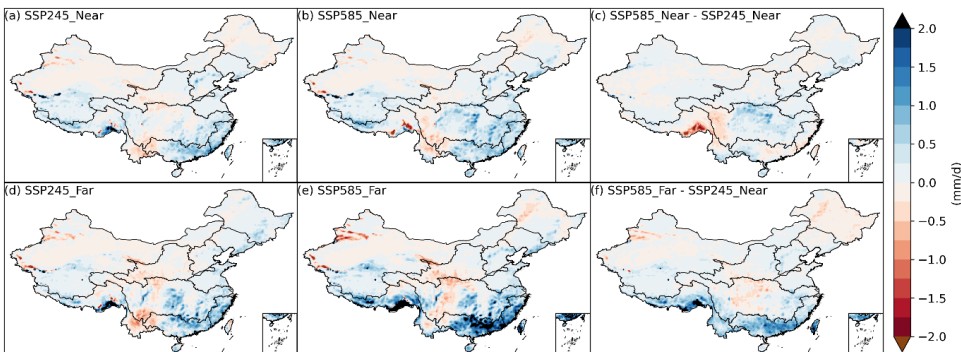

Figure 14. Multi-year ensemble mean 90th percentile runoff changes in (a) 2021-2060 and (d) 2061-2100 under the SSP245, as well as (b) 2021-2060 and (e) 2061-2100 under the SSP585, relative to the historical period (1975-2014). (c) and (f) is the difference in multi-year ensemble mean 90th percentile runoff depth between SSP245 and SSP585 in 2021-2060 and 2061-2100, respectively.

The multi-year changes in the 10th percentile runoff are illustrated in Fig. 15. In both SSP245 and SSP585 scenarios and in the near and the far futures, a decrease in the 10th percentile runoff is expected in central and southern China, with a more significant decline expected in the central Yangtze River basin, particularly under SSP585 in the far future (Fig. 15e). In the near future, only a small portion of the northeastern, southwestern, and southeastern regions are projected to experience a reduction in the 10th percentile runoff under SSP585 compared to SSP245, while in the far future, a decrease is also expected in the central Yangtze River basin. This suggests that under high emission scenarios, the central Yangtze River basin is likely to face a risk of drought.

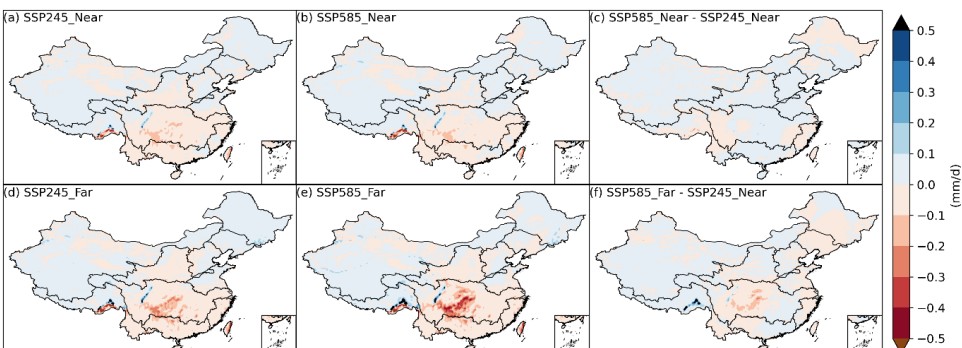


Figure 15. Multi-year ensemble mean 10th percentile runoff changes in (a) 2021-2060 and (d) 2061-2100 under the SSP245,
as well as (b) 2021-2060 and (e) 2061-2100 under the SSP585, relative to the historical period (1975-2014). (c) and (f) is the
difference in multi-year ensemble mean 10th percentile runoff depth between SSP245 and SSP585 in 2021-2060 and 2061-

2100, respectively.

**4 Discussion**
4.1 Comparisons of runoff estimates in different studies

The change trends in annual runoff depth over China are similar to the results in Zhou et al. (2023a),

indicating an overall wavelike rise, with the upward trend under SSP585 expected to be more severe than
that under SSP245. However, the rise in runoff depth under SSP245 in this study does not pass the
significance test in the Mann-Kendall trend test. In Guan et al. (2021), the increase in runoff in ten typical
basins in China under SSP585 is not consistently greater than that under SSP245. This is attributed to Guan
et al. (2021) using the climate elasticity method to project future runoff, which ignore complex hydrological
and ecological processes.

The magnitude of simulated runoff depth in this study is larger than that in Zhou et al. (2023a). On one

hand, it is mainly because the GCMs downscaled and historical hydrological modelling in this study are
based on ERA5. ERA5 generally overestimates precipitation in the northern and western regions of China,
even though it can capture seasonal variations and the broad spatial distributions in both magnitudes and





trends (Sun et al., 2021; Zhou et al., 2023c). On the other hand, the difference of simulated runoff depth may
be caused by using different models and parameterization schemes.

The spatial variations of projected runoff in this study are similar to those in other studies (Cook et

al.,2020; Wang et al., 2022; Zhou et al., 2023a). However, Cook et al. (2020) and Wang et al. (2022) analysed
runoff change by percentage change, which cannot visually convey the actual changes in runoff volume. The
percentage change in runoff is expected to be the largest in northern China, which could mislead readers into
thinking that northern China is projected to face the most dramatic change in absolute runoff volume.
However, the combined volume of runoff from six northern river basins, covering a total catchment area of
2.27 million km$^2$, contributes to less than 20% of the national total runoff. In contrast, four southern river
basins, spanning a total catchment area of 2.86 million km$^2$, contribute to over 80% of the national total
runoff (Zhang et al., 2011; Yang et al., 2022). Additionally, the runoff analysis in Cook et al. (2020) and
Wang et al. (2022) were based on global coarse resolution and did not focus on the changes within China.
Seasonal changes and changes in extreme runoff were not included in these studies (Cook et al.,2020; Wang
et al., 2022; Zhou et al., 2023a).
4.2 Comparisons of extreme runoff in different studies
For extreme runoff, the drought risk in the central Yangtze River basin is projected to be the most severe and
is expected to increase further in the far future compared to the near future, which aligns with the findings
regarding projected hydrological drought changes in the severity reported (Gu et al., (2020). Regarding
flooding, the relative change results of 100-year and 20-year flood quantiles in some GCMs indicated greater
changes in eastern and southern China river basins (Gu et al., 2021), which are consistent with the results of
this study. However, the drought and flooding analysis conducted by Gu et al. (2020, 2021) was performed
under RCP8.5 in CMIP5, and focused on specific basins of China, rather than covering the entire country.



### 4.3 The dominant driving forces for runoff changes


Runoff changes in the future under climate change primarily stem from alterations in precipitation
patterns, temperature variations, shifts in the hydrological cycle, and changes in the land surface. Continuous
global warming is expected to increase the variability of water cycle, leading to more global monsoon
precipitation, as well as the occurrence of very wet and very dry weather, climate events and seasons (IPCC,
2023). Specifically, in a warming climate, the water vapor holding capacity increases according to the
Clausius-Clapeyron law (Clapeyron, 1834; Clausius, 1850). This results in more precipitable water and
intensified precipitation extremes, which may cause flooding events. Warmer temperatures can enhance
water evaporation from the ground. As soils desiccate, the overlying air may heat up further, intensifying
evaporation and exacerbating drought conditions.
Precipitation patterns are influenced by the positions of tropical cyclones and extra-tropical cyclones
shifting poleward, which could cause drought in some regions while leading to increasing flooding events
in others (Zhang and Wang, 2017; Priestley and Catto, 2022).For perspective of the land surface, changes in
vegetation response to rising $CO_2$ levels, coupled with modifications in vegetation cover and soil moisture
in response to radiative climate change, are key contributors to projected increases in runoff (Zhou et al.,
2023b).

### 4.4 Uncertainties of the study


Due to the difficulty in obtaining gauge discharge data in China (Lin et al., 2023), we utilized limited
observational data to calibrate and validate the JULES model. Incorporating more site data distributed across
various regions of China may improve the simulation performance of the model.
Additionally, this study did not consider the influence of hydraulic engineering on runoff, which could
potentially alter the distribution of runoff and the occurrence of floods. Future research could involve





integrating data on dams, reservoirs, and other hydraulic structures into hydrological models to assess their
effects on runoff dynamics. This approach could investigate how human activities impact hydrological
processes and contribute to flood vulnerability.

The land surface model and precipitation data products introduce uncertainties into runoff extremes.

These uncertainties may increase during the propagation through models when projecting runoff extremes
in southeast China, but decreased in north China (Marthews et al., 2020).

GCMs also introduce uncertainty into hydrological modelling, and the selection of GCMs can

significantly affect the climate change impacts on hydrology (Her et al., 2019). Therefore, in this study, three
GCMs that are deemed more suitable for China were selected based on their precipitation downscaling
performance among the six GCMs evaluated. While using and screening more GCMs for hydrological
simulation may help reduce uncertainty, it also necessitates substantial computing resources.

**5 Conclusion**

In this study, we constructed a JULES model configuration specifically tailored for simulating

hydrological processes in China and employed the BCSD method to downscale and bias correct the three
selected GCMs. Using the GCMs to drive the JULES model, the future hydrological processes under medium
and high emission scenarios were projected. The main findings are summarized below:

(1)  The JULES model performed well in simulating hydrological processes in China at 0.25°

resolution. The BCSD method can effectively reduce the bias between GCMs and ERA5 in China. There are
minimal differences between downscaled-GCM-driven and ERA5-driven runoff using the JULES model
across most of China. An overestimation of runoff is shown in the southeast region, particularly pronounced
during summer months.



(2) Runoff variations across China are projected to increase significantly under the high emission
scenario, with the runoff depth increasing by 7.30 mm per decade from 2015 to 2100. Regional analysis
suggests that eastern and southern basins, notably the Southeast basin, are expected to experience the most
significant increases in runoff. Seasonal runoff trends indicate an overall increase, particularly in spring,
summer, and autumn under the high emission scenario, with varying trends observed across different
watersheds. Notably, variations in runoff trends between northern and southern China suggest contrasting
seasonal patterns, with wetter summers and drier winters expected in the south under the high emission
scenario, while the opposite trend is expected in the north.
(3) An increase in runoff across most months in the future, compared to the historical period, is
particularly evident in summer and expected to intensify in the far future under SSP585. Wetter conditions
in the near future and drier summers in the far future under SSP585 are expected in northern China.
Additionally, changes in multi-year monthly runoff patterns reveal regional variations, with some basins
projected to become drier and then wetter under SSP245, while significant increases are expected in southern
China in the far future under SSP585. Moreover, shifts from drier to wetter conditions are expected in the
southeast and southwest areas, while the middle Yangtze River basin may experience the opposite trend.
(4) Flood risk during spring, summer, and autumn may increase in the future, particularly under the
high emission scenario, while the drought risk is likely to increase in winter, summer, and autumn. Spatial
changes in the multi-year 90th percentile runoff indicate future flood risks are expected to rise in southern
China, especially in the Southwest basin, Southeast basin, Pearl River basin, and southern Yangtze River
basin, particularly under the high emission scenario in the far future. Conversely, decreases in the 10th
percentile runoff suggest a heightened risk of drought in central and southern China, with the central Yangtze
River basin facing significant declines, particularly under the high emission scenario in the far future. These



findings highlight the influence of different emission scenarios on flood and drought risks, it is important to
take proactive measures to enhance climate adaptations in the future.
**Acknowledgements**
This work was supported by the China Scholarship Council. Danyang Gao received additional support
through the visiting scientist program under the Hydro-JULES scheme at UK Centre for Ecology &
Hydrology (UKCEH) in November 2022. We acknowledge the World Climate Research Programme, which,
through its Working Group on Coupled Modelling, coordinated and promoted CMIP6. We thank the climate
modelling groups for producing and making available their model output, the Earth System Grid Federation
(ESGF) for archiving the data and providing access, and the multiple funding agencies who support CMIP6
and ESGF. We also acknowledge the use of ERA5 data produced by ECMWF. The running of JULES and
analysis of results in this work were performed on the JASMIN super-data-cluster (Lawrence et al., 2012).
JASMIN is managed and delivered by the UK Science and Technology Facilities Council (STFC) Centre for
Environmental Data Archival (CEDA).
**Data availability**
The data and code that support the study are available from the corresponding author upon request.
**Author contribution**
DG: methodology, modelling, formal analysis, and writing - original draft. AC: supervision, methodology,
and writing - review and editing. TM: supervision, modelling, and writing – review and editing. FM:
supervision, methodology, and writing – review and editing.
**Competing interests**
The contact author has declared that none of the authors has any competing interests.





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
