# Peer review of "Evaluating future hydrological changes in China under climate change"

_Hydrology and Earth System Sciences, 2024_

## Referee Comment (RC1)

This manuscript presents an in-depth study on projecting future hydrological changes in China using the Joint UK Land Environment Simulator (JULES). The study focuses on high-resolution simulations (0.25°) to assess runoff variability and flood and drought risks under medium (SSP245) and high (SSP585) emission scenarios. The results indicate significant regional variations in runoff, with wetter conditions projected in eastern and southern basins and contrasting seasonal patterns between northern and southern China. Overall, the manuscript is well written and organized, and it makes a significant contribution to understanding future hydrological changes in China. However, it should address the limitations below to enhance the reliability and practical relevance of the findings.

**Major comments:**

1.The study relies on a limited set of observational data for model calibration and validation, which may affect the accuracy and reliability of the simulations. Accurate calibration and validation are critical for ensuring the reliability of hydrological models. The limited data might lead to significant errors in the projections, undermining the study's overall credibility.

2. The influence of hydraulic engineering structures, such as dams and reservoirs, on runoff patterns is not considered, potentially overlooking significant anthropogenic impacts on hydrological processes. Human activities significantly alter hydrological processes. Ignoring these factors can lead to unrealistic projections and misinform policy decisions regarding water management and infrastructure planning.

3. The study uses three selected GCMs deemed suitable for China, but a broader range of models might provide a more comprehensive assessment of future hydrological changes and reduce model selection bias. While the selected GCMs are suitable, including a wider range of models would enhance the robustness of the projections and provide a more comprehensive picture of potential hydrological changes.

4. The ERA5 dataset, known to overestimate precipitation in some regions of China, is used for downscaling, potentially leading to inaccuracies in runoff projections. The use of ERA5 data may lead to overestimated runoff, especially in northern and western regions. This can affect the accuracy of flood and drought risk assessments.

5. The manuscript lacks a detailed discussion on the practical implications of the findings for water resource management and climate adaptation strategies in China. While the scientific findings are robust, their practical application in policy and management is crucial. A more detailed discussion could help translate the scientific results into actionable strategies for stakeholders.

**Minor comments:**

1. Line 61-62,please specify how coarse it is.

2. Line 83, it should be Section 2.1.

3. In figure 1, it is recommended to add 10 dashed lines at present.

4. I would recommend to add the control area for the selected gauges somewhere.

---

## Author Comment (AC1)

**Reviewers' comments in blue. Our responses in black. Yellow highlighting emphasises revision undertaken.**

**Reviewer 1:**

This manuscript presents an in-depth study on projecting future hydrological changes in China using the Joint UK Land Environment Simulator (JULES). The study focuses on high-resolution simulations (0.25°) to assess runoff variability and flood and drought risks under medium (SSP245) and high (SSP585) emission scenarios. The results indicate significant regional variations in runoff, with wetter conditions projected in eastern and southern basins and contrasting seasonal patterns between northern and southern China. Overall, the manuscript is well written and organized, and it makes a significant contribution to understanding future hydrological changes in China. However, it should address the limitations below to enhance the reliability and practical relevance of the findings.

Thank you for taking your time to review our manuscript.

Major comments:

1.The study relies on a limited set of observational data for model calibration and validation, which may affect the accuracy and reliability of the simulations. Accurate calibration and validation are critical for ensuring the reliability of hydrological models. The limited data might lead to significant errors in the projections, undermining the study's overall credibility.

Thank you for your valuable feedback. We acknowledge that the quantity and quality of observational data are critical for accurate calibration and validation of hydrological models. However, it remains difficult to access and use observed discharge data in China (Lin et al., 2023). We used the best available observational data to calibrate and validate our model. While the limited dataset may introduce some level of uncertainty, we conducted validation at multiple sites to assess the model performance. Although the number of sites is limited, the results indicate that the model performs acceptably within the available dataset.

We agree that more extensive observational data would improve the accuracy and reliability of the model. We have already included the discussion about discharge data in Lines 365-367.

We plan to add more discussion in Section 4.4 as following: Due to the difficulty in obtaining gauge discharge data in China (Lin et al., 2023), we used limited observational data to calibrate and validate the JULES model. Although the number of sites is limited, the results indicate that the model performs acceptably within the available dataset. If more site data distributed across various regions of China can be obtained and applied to calibration and validation, the model performance could be further improved.

Lin, J., Bryan, B. A., Zhou, X., Lin, P., Do, H. X., Gao, L., Gu, X., Liu, Z., Wan, L., Tong, S., Huang, J., Wang, Q., Zhang, Y., Gao, H., Yin, J., Chen, Z., Duan, W., Xie, Z., Cui, T., … Yang, Z. (2023). Making China's water data accessible, usable and shareable. Nature Water, 1(4), Article 4. https://doi.org/10.1038/s44221-023-00039-y

2. The influence of hydraulic engineering structures, such as dams and reservoirs, on runoff patterns is not considered, potentially overlooking significant anthropogenic impacts on hydrological processes. Human activities significantly alter hydrological processes. Ignoring these factors can lead to unrealistic projections and misinform policy decisions regarding water management and infrastructure planning.

Thank you for your comments. Our study primarily focuses on an understanding of the impacts of climate change on hydrological processes. Investigating how hydraulic structures impact hydrological processes is beyond our scope. Consequently, we did not incorporate the effects of hydraulic engineering structures into our model.

We agree that hydraulic engineering structures have effects on runoff patterns. We have already included it in Lines 368-372.

We plan to enhance Section 4.4 as following: Additionally, this study did not consider the influence of hydraulic engineering on runoff, which could potentially alter the rainfall-runoff response. Our study primarily focuses on understanding the impacts of climate change on hydrological processes. Investigating how hydraulic structures affect such processes is beyond our scope. Consequently, we did not incorporate the effects of hydraulic engineering structures into our model. Future research could involve integrating data on dams, reservoirs, and other hydraulic structures into hydrological models to assess their effects on runoff dynamics. This approach could investigate how human activities impact hydrological processes and contribute to flood vulnerability.

3. The study uses three selected GCMs deemed suitable for China, but a broader range of models might provide a more comprehensive assessment of future hydrological changes and reduce model selection bias. While the selected GCMs are suitable, including a wider range of models would enhance the robustness of the projections and provide a more comprehensive picture of potential hydrological changes.

Thank you for your comments. We chose six CMIP6 GCMs based on their performance in representing regional climate variability and their availability in high temporal resolution suitable for our hydrological model. While using more GCMs can indeed provide a broader range of uncertainties, we need to make a trade-off between the number of GCMs and the computational resources required. The ideal situation is to select a few models that can represent the majority of the GCMs and have a small bias from the observations.

Our first step in GCM selection was to identify six GCMs based on previous studies that demonstrated their good performance for precipitation and temperature in China (Yang et al., 2021; Lu et al., 2022; Jia et al., 2023). Next, we downscaled precipitation from these six GCMs and compared the bias with ERA5 datasets. Therefore, the GCMs we selected have biases that are as small as possible when compared to the observed data. These steps have been already included in the manuscript (Section2.2, Lines 128-137).

We will revise our manuscript to include evidence that the GCMs we selected can represent the majority of the GCMs. We downloaded 19 models (including ACCESS-CM2, ACCESS-ESM1-5, CanESM5, CMCC-ESM2, CNRM-ESM2-1, EC-Earth3, EC-Earth3-Veg, FGOALS-g3, GFDL-ESM4, INM-CM4-8, INM-CM5-0, MIROC-ES2L, MPI-ESM1-2-HR, MPI-ESM1-2-LR, MRI-ESM2-0,

NorESM2-LM, NorESM2-MM, TaiESM1, and UKESM1-0-LL) according to the list in NEX-GDDP-CMIP6 (Thrasher et al., 2022) and calculated the daily average precipitation and temperature from 1959 to 2014 of all GCM ensemble means and selected GCM ensemble means. The selected GCMs mean temperature is 6.07 °C while the mean for all GCMs is 6.03 °C. The selected GCMs mean precipitation is 2.08 mm/day while the mean for all GCMs is 2.45 mm/day. Therefore, the GCMs we selected are representative of the performance of most GCMs.

To present this evidence, we plan to include the following text in Section 2.2: To ensure the selected GCMs represent the performance of most GCMs, daily average precipitation and temperature from 1959 to 2014 for the selected GCM ensemble means were compared with the ensemble means of 19 GCMs. These GCMs include ACCESS-CM2, ACCESS-ESM1-5, CanESM5, CMCC-ESM2, CNRM-ESM2-1, EC-Earth3, EC-Earth3-Veg, FGOALS-g3, GFDL-ESM4, INM-CM4-8, INM-CM5-0, MIROC-ES2L, MPI-ESM1-2-HR, MPI-ESM1-2-LR, MRI-ESM2-0, NorESM2-LM, NorESM2-MM, TaiESM1, and UKESM1-0-LL.

And we plan to include the following text in Section 3.2: The selected GCMs mean temperature is 6.07 °C while the mean for all 19 GCMs is 6.03 °C. The selected GCMs mean precipitation is 2.08 mm/day while the mean for all 19 GCMs is 2.45 mm/day. Therefore, the GCMs we selected are representative of the performance of most GCMs.

Jia, Q., Jia, H., Li, Y., & Yin, D. (2023). Applicability of CMIP5 and CMIP6 Models in China: Reproducibility of Historical Simulation and Uncertainty of Future Projection. Journal of Climate, 36(17), 5809–5824. https://doi.org/10.1175/JCLI-D-22-0375.1

Lu, K., Arshad, M., Ma, X., Ullah, I., Wang, J., & Shao, W. (2022). Evaluating observed and future spatiotemporal changes in precipitation and temperature across China based on CMIP6-GCMs. International Journal of Climatology, 42(15), 7703–7729. https://doi.org/10.1002/joc.7673

Yang, X., Zhou, B., Xu, Y., & Han, Z. (2021). CMIP6 Evaluation and Projection of Temperature and Precipitation over China. Advances in Atmospheric Sciences, 38(5), 817–830. https://doi.org/10.1007/s00376-021-0351-4

4. The ERA5 dataset, known to overestimate precipitation in some regions of China, is used for downscaling, potentially leading to inaccuracies in runoff projections. The use of ERA5 data may lead to overestimated runoff, especially in northern and western regions. This can affect the accuracy of flood and drought risk assessments.

Thank you for your insightful comments. ERA5 generally overestimates precipitation in the northern and western regions of China, but it can capture seasonal variations and the broad spatial distributions in both magnitudes and trends (Sun et al., 2021; Zhou et al., 2023). We have already included this information in the discussion section in the manuscript (Section 4.1, Lines 324-326).

We used three meteorological datasets to drive the model, ERA5 performs best among them. We agree with other scholars that, despite its biases, ERA5 remains one of the best reanalysis datasets available, providing a comprehensive set of variables (Cucchi et al., 2020; Xu et al.,

2021). ==We plan to include this information in Section 4.1:== We used three meteorological datasets to drive the model, including ERA5, Global Meteorological Forcing Dataset (GMFD, Sheffield et al., 2006) and China Meteorological Forcing Dataset (CMFD, He et al., 2020). Our evaluation showed that ERA5 performed the best compared to the other datasets. We agree with other scholars that, despite its biases, ERA5 remains one of the best reanalysis datasets available, providing a comprehensive set of variables (Cucchi et al., 2020; Xu et al., 2021).

Cucchi, M., Weedon, G. P., Amici, A., Bellouin, N., Lange, S., Müller Schmied, H., Hersbach, H., & Buontempo, C. (2020). WFDE5: Bias-adjusted ERA5 reanalysis data for impact studies. Earth System Science Data, 12(3), 2097–2120. https://doi.org/10.5194/essd-12-2097-2020

He, J., Yang, K., Tang, W., Lu, H., Qin, J., Chen, Y., & Li, X. (2020). The first high-resolution meteorological forcing dataset for land process studies over China. Scientific Data, 7(1), 25. https://doi.org/10.1038/s41597-020-0369-y

Sheffield, J., Goteti, G., & Wood, E. F. (2006). Development of a 50-Year High-Resolution Global Dataset of Meteorological Forcings for Land Surface Modeling. Journal of Climate. https://doi.org/10.1175/JCLI3790.1

Sun, H., Su, F., Yao, T., He, Z., Tang, G., Huang, J., Zheng, B., Meng, F., Ou, T., & Chen, D. (2021). General overestimation of ERA5 precipitation in flow simulations for High Mountain Asia basins. Environmental Research Communications, 3(12), 121003. https://doi.org/10.1088/2515-7620/ac40f0

Xu, Z., Han, Y., Tam, C.-Y., Yang, Z.-L., & Fu, C. (2021). Bias-corrected CMIP6 global dataset for dynamical downscaling of the historical and future climate (1979–2100). Scientific Data, 8(1), 293. https://doi.org/10.1038/s41597-021-01079-3

Zhou, Z., Chen, S., Li, Z., & Luo, Y. (2023c). An Evaluation of CRA40 and ERA5 Precipitation Products over China. Remote Sensing, 15(22), Article 22. https://doi.org/10.3390/rs15225300

5. The manuscript lacks a detailed discussion on the practical implications of the findings for water resource management and climate adaptation strategies in China. While the scientific findings are robust, their practical application in policy and management is crucial. A more detailed discussion could help translate the scientific results into actionable strategies for stakeholders.

Thank you for your suggestions. ==We plan to add a Section 4.5 to enhance the discussion section of our manuscript:==

4.5 Practical implications

Given the projected increase in runoff depth by 7.30 mm per decade under the high emission scenario, water resource managers should prepare for higher water availability, especially in eastern and southern basins. This information is vital for optimising water storage and distribution systems to prevent waste and ensure equitable water distribution.

The expected increase in runoff during spring, summer, and autumn under high emission

scenarios necessitates seasonal adaptation measures. For instance, enhanced flood control infrastructure will be essential in the southeast basin to mitigate the heightened flood risk.

The projection of drier winters in southern China and contrasting seasonal patterns between northern and southern regions highlight the need for region-specific drought preparedness strategies. Investments in drought-resistant crops and efficient irrigation systems will be crucial in northern China.

Minor comments:

1. Line 61-62, please specify how coarse it is.

Thanks for your suggestion. We plan to add the resolution information in the sentence in Lines 59-62 as following: However, their analysis mainly based on results from CMIP5, CMIP6, Inter-Sectoral Impact Model Inter-Comparison Project (ISMIP2a) and Global Land Data Assimilation System (GLDAS), the resolutions of their runoff projections were coarse (≥0.5°).

2. Line 83, it should be Section 2.1.

Thanks for your careful review. We will revise it.

3. In figure 1, it is recommended to add 10 dashed lines at present.

Thanks for your suggestion. We will update the 9 dashed lines used currently to 10 dashed lines.

4. I would recommend to add the control area for the selected gauges somewhere.

Thanks for your suggestion. We have carefully reviewed the GRDC dataset, and unfortunately, it does not include explicit control area information for the gauges. However, we have provided the locations of the gauges in Fig. 1 as a reference.

---

## Author Comment (AC2)

**Reviewers' comments in blue. Our responses in black. Yellow highlighting emphasises revision undertaken.**

**Reviewer 2:**

This study applies 'the Joint UK Land Environment Simulator (JULES)' to simulate and project runoff in China at a high resolution; and further analyze the flood and drought risks. The authors claim that 1) annual runoff in China is projected to increase significantly, notably in eastern and southern basins; 2) northern China is expected to have wetter conditions in the near future and drier summers in the far future; 3) southern China is projected to face greater flood risks; 4) central Yangtze River basin can face intensified drought risk. Overall, the idea of using the JULES to project runoff in China is kind of good; However, the whole story is really boring and unclear, and there are some technique problems exist. Overall, I consider this manuscript cannot be published at such a good journal at this stage.

Thank you for taking your time to review our manuscript.

Major concerns:

1. In abstract, I cannot see any logic in reporting the results but only redundancy. The authors only say e.g., annual runoff increase; southern China runoff increase…; all of them is only qualitative description without a single robust number to show the results; besides, no mechanisms at all to reveal why the runoff is projected to change like that. Because of increasing precipitation? Evapotranspiration? This makes the whole story really boring! And I cannot even see a clear storyline to show how the spatial patterns of runoff changes in China!

Thanks for your comments. The abstract mainly presents the key messages from the regions showing significant shift. Granular details are presented in the paper. We plan to revise the abstract to include a more structured presentation of our findings and important numbers as following: Projecting and understanding future hydrological changes in China are critical for effective water resource management and adaptation planning in response to climate variability. However, few studies have investigated runoff variability, as well as flood and drought risks under climate change scenarios for the entire region of China at high resolution. In this study, we use the Joint UK Land Environment Simulator (JULES), specifically tailored for simulating hydrological processes in China at a 0.25-degree resolution. The model is driven by downscaled and bias-corrected data from Global Climate Models (GCMs), using the bias-correction and spatial disaggregation (BCSD) method, to project future hydrological processes under medium (SSP245) and high (SSP585) emission scenarios. The results indicate a significant increase in annual runoff across China under the high emission scenario, with a projected increase of 7.30 mm per decade, particularly in the eastern and southern basins. Regional patterns emerge, with wetter summers and drier winters expected in southern China, while northern China is projected to experience drier summers in the far future. Furthermore, shifts from drier to wetter conditions are projected in the southeast and southwest areas, while the middle Yangtze River basin is expected to experience the opposite trend. Flood risks are projected to rise in spring, summer, and autumn, particularly in southern China, while drought risks are expected to intensify in the central Yangtze River basin, especially in the far future. These findings highlight the influence of different emission scenarios on flood and

drought risks, emphasizing the need for proactive measures to enhance climate adaptation in the future.

The main causes of runoff change include natural factors such as climate change, and anthropogenic factors such as land use change, water conservancy projects (Zhai and Tao, 2017). Our purpose is to investigate the runoff change under climate change, without consideration of anthropogenic factors. Therefore, the runoff changes discussed in this research are primarily driven by natural variations in the water cycle. We have already included the discussion about the dominant driving forces for runoff changes in Section 4.3, Lines 348-363.

Zhai, R., & Tao, F. (2017). Contributions of climate change and human activities to runoff change in seven typical catchments across China. *Science of The Total Environment*, *605–606*, 219–229. https://doi.org/10.1016/j.scitotenv.2017.06.210

2. Line 116: a reference period from 1959-2014 were chosen to generate two CDFs. This reference period contains 65 years in total and involves significant warming and is not stationary!! How can the authors choose such a long time period as a reference period? The typical time period only involves 20-30 years to ensure the stationarity of the data!

Thank you for your question regarding the length of the reference period. We have followed a similar approach to the bias correction and spatial disaggregation (BCSD) method used in the NASA Earth Exchange Global Daily Downscaled Projections (NEX-GDDP-CMIP6, Thrasher et al., 2022) They used reference period of 1960-2014 as the basis for the cumulative distribution function (CDF).

A longer reference period can provide a more robust dataset by encompassing a wider range of climate variability and extreme events, which is beneficial for statistical modelling. In our study, we have chosen a reference period from 1959 to 2014, ensuring it is long enough to capture significant climate trends while maintaining relevance to current and future climate conditions.

Thrasher, B., Wang, W., Michaelis, A. et al. NASA Global Daily Downscaled Projections, CMIP6. Sci Data 9, 262 (2022). https://doi.org/10.1038/s41597-022-01393-4

3. The authors only use six CMIP6 GCMs to downscale and simulate and project river runoff. I fully cannot understand why they need to downscale themselves? The ISIMIP 3b already provide high-resolution downscale climate output (https://www.isimip.org/newsletter/isimip3a3b-protocol-published/), they also use ERA5 as the reference dataset. And I consider their bias correction methodology is even better than the authors did. If the authors use more than 15 models to specifically consider the uncertainty and associated source, then it is fine; But the authors only use 6 models while ISIMIP3b provide 5 model outputs with really good bias correction, then why the authors perform bias correction themselves? If they do, I need to see the superiority of their bias correction results comparing to the ISIMIP 3b. (The ISIMIP 3b is a publicly accessible dataset and without comparison, I would like to trust them more comparing the authors did).

Thank you for your questions. We appreciate the opportunity to clarify our methodology and

the rationale behind our choices.

**Selection of CMIP6 GCMs:**

We chose six CMIP6 GCMs based on their performance in representing regional climate variability and their availability in high temporal resolution suitable for our hydrological model. While using more GCMs can indeed provide a broader range of uncertainties, we need to make a trade-off between the number of GCMs and the computational resources required. The ideal situation is to select a few models that can represent the majority of the GCMs and have a small bias from the observations.

Our first step in GCM selection was to identify six GCMs based on previous studies that demonstrated their good performance for precipitation and temperature in China (Yang et al., 2021; Lu et al., 2022; Jia et al., 2023). Next, we downscaled precipitation from these six GCMs and compared the bias with ERA5 datasets. Therefore, the GCMs we selected have biases that are as small as possible when compared to the observed data. These steps have been already included in the manuscript (Section 2.2, Lines 128-137).

We will revise our manuscript to include evidence that the GCMs we selected can represent the majority of the GCMs. We downloaded 19 models (including ACCESS-CM2, ACCESS-ESM1-5, CanESM5, CMCC-ESM2, CNRM-ESM2-1, EC-Earth3, EC-Earth3-Veg, FGOALS-g3, GFDL-ESM4, INM-CM4-8, INM-CM5-0, MIROC-ES2L, MPI-ESM1-2-HR, MPI-ESM1-2-LR, MRI-ESM2-0, NorESM2-LM, NorESM2-MM, TaiESM1, and UKESM1-0-LL) according to the list in NEX-GDDP-CMIP6 (Thrasher et al., 2022) and calculated the daily average precipitation and temperature from 1959 to 2014 of all GCM ensemble means and selected GCM ensemble means. The selected GCMs mean temperature is 6.07 °C while the mean for all GCMs is 6.03 °C. The selected GCMs mean precipitation is 2.08 mm/day while the mean for all GCMs is 2.45 mm/day. Therefore, the GCMs we selected are representative of the performance of most GCMs.

To present this evidence, we plan to include the following text in Section 2.2: To ensure the selected GCMs represent the performance of most GCMs, daily average precipitation and temperature from 1959 to 2014 for the selected GCM ensemble means were compared with the ensemble means of 19 GCMs. These GCMs include ACCESS-CM2, ACCESS-ESM1-5, CanESM5, CMCC-ESM2, CNRM-ESM2-1, EC-Earth3, EC-Earth3-Veg, FGOALS-g3, GFDL-ESM4, INM-CM4-8, INM-CM5-0, MIROC-ES2L, MPI-ESM1-2-HR, MPI-ESM1-2-LR, MRI-ESM2-0, NorESM2-LM, NorESM2-MM, TaiESM1, and UKESM1-0-LL. And we plan to include the following text in Section 3.2: The selected GCMs mean temperature is 6.07 °C while the mean for all 19 GCMs is 6.03 °C. The selected GCMs mean precipitation is 2.08 mm/day while the mean for all 19 GCMs is 2.45 mm/day. Therefore, the GCMs we selected are representative of the performance of most GCMs.

**Why Downscale by Ourselves:**

The decision to perform our own downscaling was driven by the specific needs of our study region and objectives. Although ISIMIP3b provides downscaled climate outputs, their resolution (0.5°) does not meet our requirements. The resolution provided by ISIMIP3b, while excellent for broader applications, does not capture the finer-scale variability required for our

hydrological modelling. Additionally, ISIMIP3b does not include the SSP245 scenario. Our study aims to analyse runoff changes at a finer resolution (0.25°) under medium (SSP245) and high (SSP585) emission scenarios, which necessitated the use of our own downscaling approach.

NEX-GDDP-CMIP6 (Thrasher et al., 2022) provides driving data at 0.25° except for surface pressure. They downscaled CMIP6 based on the Global Meteorological Forcing Dataset (GMFD). For consistency of input data, historical modelling should use GMFD datasets if we want to use NEX-GDDP-CMIP6 data. However, we found that using GMFD datasets for calibration and validation of the JULES model did not perform as well as using ERA5 in China. Therefore, we decided to downscale ourselves based on ERA5 datasets.

The bias correction methodology we employed is based on the Bias Correction and Spatial Disaggregation (BCSD) method, which is a well-established approach used in various studies (Wood et al., 2004; Thrasher et al., 2022). This method has been shown to effectively reduce biases in climate projections. While we acknowledge the robustness of the ISIMIP3b bias correction, our methodology is specifically tailored to our regional study area and includes a detailed comparison with ERA5 data to ensure accuracy. The performance of the downscaled GCMs compared with ERA5 has already been included in Section 3.2, Lines 167-199 in the manuscript and Fig. S1, S2 and S3 in the supplementary.

Thrasher, B., Wang, W., Michaelis, A. et al. NASA Global Daily Downscaled Projections, CMIP6. Sci Data 9, 262 (2022). https://doi.org/10.1038/s41597-022-01393-4

Jia, Q., Jia, H., Li, Y., & Yin, D. (2023). Applicability of CMIP5 and CMIP6 Models in China: Reproducibility of Historical Simulation and Uncertainty of Future Projection. Journal of Climate, 36(17), 5809–5824. https://doi.org/10.1175/JCLI-D-22-0375.1

Lu, K., Arshad, M., Ma, X., Ullah, I., Wang, J., & Shao, W. (2022). Evaluating observed and future spatiotemporal changes in precipitation and temperature across China based on CMIP6-GCMs. International Journal of Climatology, 42(15), 7703–7729. https://doi.org/10.1002/joc.7673

Wood, A. W., Leung, L. R., Sridhar, V., & Lettenmaier, D. P. (2004). Hydrologic Implications of Dynamical and Statistical Approaches to Downscaling Climate Model Outputs. Climatic Change, 62(1–3), 189–216. https://doi.org/10.1023/B:CLIM.0000013685.99609.9e

Yang, X., Zhou, B., Xu, Y., & Han, Z. (2021). CMIP6 Evaluation and Projection of Temperature and Precipitation over China. Advances in Atmospheric Sciences, 38(5), 817–830. https://doi.org/10.1007/s00376-021-0351-4

4. In Lines 157-159: there is a sharp decrease with regard to the hydrological model performance. Is it because of over-fitting? Or what reasons cause this? Why the authors do-not use cross-validation (e.g., Arsenault et al., 2018) to calibrate the hydrological models?

Arsenault, R., Brissette, F., & Martel, J. L. (2018). The hazards of split-sample validation in hydrological model calibration. Journal of hydrology, 566, 346-362.

Thank you for pointing this out. We apologize for any confusion caused by our wording. In

Lines 157-159, we stated: "The r and NSE values are above 0.83 and 0.64, respectively, for calibrations, and greater than 0.78 and 0.58, respectively, for validation, indicating that the simulation outcomes are acceptable." The performance metrics (r and NSE) during the calibration and validation periods vary by station. At some stations, the r and NSE values are higher during calibration than validation, while at other stations, the opposite is true. Therefore, there is no sharp decrease in hydrological model performance. We only described the minimum values, which might have been misleading. We plan to revise this sentence to clarify: The r and NSE values in monthly calibrations and validations are greater than 0.78 and 0.58, respectively, indicating that the simulation outcomes are acceptable.

Regarding cross-validation, it is widely used in machine learning due to its ability to improve model performance and prevent over-fitting through multiple training and verification iterations. With advances in computing resources, the training speed of machine learning models has increased significantly, making it feasible to perform these multiple iterations in a reasonable time.

In your mentioned paper, Arsenault et al. (2018) proposed a cross-validation method for hydrological models, specifically using the GR4J-CN (a 6-parameter lumped model) and HMETS (a lumped-conceptual model). These lumped hydrological models have simpler structures and typically require less computationally intensive driving data, such as rainfall and temperature. Therefore, similar to machine learning models, these lumped hydrological models can be calibrated and validated multiple times in a reasonable timeframe.

However, for more complex and longer-running models like the land surface model (JULES) we used, cross-validation is not practical due to their complex processes and significant computational resource requirements. As Arsenault et al. (2018) noted, "There is a trend in hydrological science toward increasingly more complex process-based and/or distributed hydrological models with some models now implementing land surface schemes with complex formulations such as the Richards nonlinear differential equations of water movement in the non-saturated portion of the soil column. In such models, calibration over the full dataset length would not be possible without access to massive parallel computing facility."

Therefore, while cross-validation is a valuable tool in many contexts, its application in highly complex and resource-intensive models like JULES is limited by current computational constraints.

5. In Lines 160-162: 'Most stations with good performance are large rivers, indicating that the model simulates better in large rivers. This is really unreasonable! Large rivers involve reservoirs and hydraulic engineering, land cover and land type changes, which can evidently affect runoff regimes and is more difficult to simulate, irrespective of model resolution. This is why most previous studies focus on catchment scale and especially small catchments. The JULES model also does not consider land use and land cover change, nor the impacts from reservoirs, how can the performance be better in large catchments than small ones?

Thank you for your question. Our comparison is the relative comparison between large and small rivers to highlight the differences in model performance across different scales. Our simulation is on a resolution of 0.25°, which means that within a 0.25° x 0.25° grid, we use

simulated data from the point closest to the station among the four corners of the grid to compare with the station data. For large rivers, the selected point is very likely to be within the same river as the station, whereas for small rivers, sometimes the conditions at the point can be different from the station's location.

We appreciate that an inclusion of made features including reservoirs and large rivers representation can further improve predictions. We have already included it in Lines 368-372.

We plan to enhance Section 4.4 as following: Additionally, this study did not consider the influence of hydraulic engineering on runoff, which could potentially alter the rainfall-runoff response. Our study primarily focuses on understanding the impacts of climate change on hydrological processes. Investigating how hydraulic structures affect such processes is beyond our scope. Consequently, we did not incorporate the effects of hydraulic engineering structures into our model. Future research could involve integrating data on dams, reservoirs, and other hydraulic structures into hydrological models to assess their effects on runoff dynamics. This approach could investigate how human activities impact hydrological processes and contribute to flood vulnerability.

6. Most importantly! All calibration and validation of runoff are based on a monthly scale, how can the authors then use the calibrated models to simulate daily runoff (e.g., Figs. 7-8)?! I cannot trust the results in this case. If the authors need to analyze the daily runoff variations, they need to train and validate the models at the daily scale.

Thank you for your comments. While our initial calibration used monthly data due to data availability, we subsequently validated our model using daily data at four stations. This allowed us to assess model performance at a finer temporal scale. The results indicated satisfactory performance (NSE > 0.53). We plan to add the following figure and text in Section 3.1 and supplement: The r and NSE values of daily validation are greater than 0.81 and 0.53, respectively, indicating that the simulation effect on the daily scale is also acceptable (Fig. S1).

[Figure]

Figure S1. Comparison of observed and simulated discharge in daily validation

7. In 278-285: again, I consider the authors really donot know basic concepts of hydrology. 1) if using monthly data to calibrate the hydrological model, then the calibrated models cannot be used to simulate daily runoff! Nor can they be used to simulate and project floods which are measured at the daily scale! 2) the low flows, e.g., 10th percentile runoff, cannot be directly used to indicate droughts! The drought episode is typically defined as the abnormally runoff deficits persist for a long time! Not the change of low flow!

Thanks for. For daily validation, as our response to your last comment, we will add figures and detailed descriptions in Section 3.1 and supplement.

Thank you for your comments regarding drought. Our main purpose is to evaluate potential drought risks rather than to identify specific drought events. Therefore, our focus is primarily on changes in low flows, rather than on the duration of these low flow conditions.

Specific concerns:

1. Figs 2-3, please change the unit of m3/s to mm by dividing the discharge area.

Thank you. There seems to be a misunderstanding regarding the standard units used in hydrology. Discharge is typically expressed in cubic meters per second (m³/s) rather than millimetres (mm). This is a standard practice in the field. Refsgaard (1997) provides a comprehensive illustration of the steps involved in distributed hydrological modelling and correctly uses m³/s for calibration and validation. I recommend reviewing this article to gain a clearer understanding of these standard practices.

Refsgaard, J. C. (1997). Parameterisation, calibration and validation of distributed hydrological models. Journal of Hydrology, 198(1), 69–97. https://doi.org/10.1016/S0022-1694(96)03329-X

2. In all figures, the labels and characters are really too small.

Thank you. We will address these in the revised version.